# Effect of Butt Gap on Stress Distribution and Carrying Capacity of X80 Pipeline Girth Weld

**DOI:** 10.3390/ma15238299

**Published:** 2022-11-22

**Authors:** Lixia Zhu, Haidong Jia, Xiao Li, Jinheng Luo, Lifeng Li, Dongdong Bai

**Affiliations:** 1CNPC Tubular Goods Research Institute, Xi’an 710077, China; 2Pipeline Network Group (Xinjiang) United Pipeline Co., Ltd., Urumqi 830011, China; 3School of Materials Science and Engineering, Xi’an Shiyou University, Xi’an 710065, China

**Keywords:** numerical simulation, temperature field, residual stress, carrying capacity

## Abstract

An unstable assembly gap is detrimental to the formation and performance of the pipeline butt girth weld joint. Therefore, a numerical model of an 18.4 mm-thick X80 pipeline girth weld by a homogeneous body heat source was established to investigate the effect of the butt gap on the joint temperature and stress field, and carrying capacity. The accuracy of the simulation results was verified by measuring the welding thermal cycle with a thermocouple. The investigation results showed that the weld pool, heat-affected zone (HAZ) width, and maximum circumferential stress of the joint rose with the increase in the butt gap. The tensile stress unfavorable to the joint quality was mainly distributed in the weld metal and partial HAZ, and the distribution areas gradually expanded as the gap increased. The Von Mises stress peak value of the joint appeared in the order of 3 mm > 2 mm > 1 mm > 0 mm gap, reaching the maximum of 467.3 MPa (3 mm gap). This variation trend is directly related to the improvement in welding heat input with increasing butt gaps. The maximum Von Mises stress of the joint was positively correlated with the carrying capacity of the pipeline, which diminished as the butt gap enlarged. The pipeline carrying capacity reached 17.8 MPa for the joint with no butt gap, and dropped to 13.1 MPa for the joint with a 3 mm gap. The relationship between the carrying capacity (*P*) and butt gap (*C*) was described by *P* = −0.125*C*^2^ − 1.135*C* + 17.715, through which the pipeline carrying capacity with other butt gaps can be predicted.

## 1. Introduction

Welding is the primary method to manufacture metal components in industry [1,2]. Butt joints are frequently applied in oil and gas transportation, construction machinery, and pressure vessels. Abundant studies have been carried out in this aspect, in which the influence of welding energy parameters (welding current, arc voltage, welding speed, etc.) on the weld quality plays an important role. Zhang et al. studied the gas tungsten arc welding (GTAW) waveform influences on the weld formation, microstructure, and mechanical properties of AZ31B Mg alloy [3]. Sandhya et al. investigated the influence of welding speed, voltage, and beam current on the microstructure and mechanical properties of electron-beam-welded titanium radial joints [4]. Xu et al. conducted a numerical investigation on the influence of current waveform on droplet transfer in pulsed gas metal arc welding (GMAW) [5].

Besides the welding current, arc voltage, and welding speed, the assembly gap is also crucial in the butt weld, which can directly affect the welding penetration condition. Burn-through defects may occur if the butt gap is increased to a critical value even with the same welding heat input. At the production site, the reserved butt joint gap is more likely to be affected by welding deformation, fixture accuracy, and the site environment. So, it is of great significance to study the adaptability of the welding process to an assembly gap. Some researchers have studied the butt gap in joint performance. Huang et al. implemented laser-MIG welding of a low-alloy high-strength steel sheet with a 1 mm butt gap and optimized laser-wire distance [6]. Hao et al. established a three-dimensional transient numerical analysis model of tungsten inert gas (TIG) welding with a reserved gap, and analyzed the dynamic variations in the flow field and deformation in the weld pool. Their findings indicated that the liquid metal flowed from both sides to the middle, bringing heat into the gap, and the liquid metal in front of the weld pool flowed to the back of the weld pool through the gap, increasing the weld penetration [7]. Li et al. investigated variable narrow gap welding by rotating GMAW, and developed effective algorithms to extract the weld groove features [8]. Tsarkov et al. conducted an investigation of butt welding 5mm-thick sheet 5083 aluminum alloy in the presence of joint gaps, indicating that the joint efficiencies were low when the gaps exceeded 1.5 mm [9]. Nomura et al. constructed a deep learning model using a monitoring image during the welding to predict the welding quality in a single bevel GMAW with gap fluctuation. AZ31 magnesium alloy sheets that were 2mm-thick were welded by friction stir welding (FSW) in butt joint configuration using various gap widths [10]. Chiuzuli et al. found that the lack-of-fill defect occurred on the top surface when the gap width increased to 1.15 mm, and the gap width did not significantly affect the microstructure of the weld zones and their hardness values in all joints [11]. Qiang et al. conducted a comparative investigation on the weld appearance of a double-sided synchronization GTAW procedure with a butt gap and without a butt gap [12]. Wang et al. investigated the dynamic behavior of the weld pool and weld formation. They found that the weld formation of the welding direction from a large gap to a small gap was much worse than that from a small gap to a large gap in the case of the same gap change rate [13]. In addition, the influence of air gap on the glass-to-glass welding using picosecond laser technology was studied, and laser welding with good performance was achieved with an air gap of 2 μm [14].

Micro-alloyed thermomechanical rolled steel was extensively applied in the field of oil and gas transportation. Khalaj et al. established a model to predict the austenite grain size in the X70 weld heat-affected zone (HAZ) [15], and another model based on an artificial neural network to forecast the martensite fraction of micro-alloyed steels [16]. Their group also developed an approach based on the artificial neural network of predicting the ultimate tensile strength and toughness of pipeline steels [17,18].

Nowadays, X80 is the most widely used micro-alloyed pipeline steel in oil and gas transportation [19,20,21,22,23], and it is required to be welded on site. The joint form on the pipeline is mainly butt girth weld [24], and the pipe needs to be assembled with an alignment device prior to welding. Owing to relatively the low machining accuracy of pipe grooves, the alignment precision is often difficult to guarantee. So, the butt gaps of the girth weld are often inconsistent. In addition, the deformation in the welding process also intensifies the change in the gap. The gap inconsistency will seriously affect the weld formation of X80 pipeline steel, easily resulting in a lack of penetration or burn-through. Consequently, it is essential to investigate the butt gap variation in the welding of X80, which has hardly been reported.

The pipeline steel thickness is large, so the multi-pass GMAW process is commonly exploited. The temperature and stress field cannot be obtained through the experiment. In this study, the butt gap’s influences on the temperature field, stress field, and carrying capacity of 18.4 mm-thick X80 butt girth weld were investigated in detail through numerical simulation.

## 2. Materials and Methods

### 2.1. Geometric Model and Meshing

ABAQUS software V6.10 (Dassault Systems, Paris, France) was used to calculate the temperature field, stress field, and carrying capacity. The 1/8 analysis model of the pipeline was established according to the symmetry of the girth weld, as shown in Figure 1. Figure 2 illustrates the weld bead division and meshed models with diverse butt gaps. The specification of the pipe model was 1219 mm × 18.4 mm and the length was 300 mm. The assembly gap was 0 mm, 1 mm, 2 mm, and 3 mm, respectively. All grids used in the model were C3D8R, eight-node linear hexahedral elements. The weld bead and its HAZ were densely meshed, with a grid size of about 0.5–0.8 mm. The elements number of the joint were 201,468 with a gap of 0 mm, 202,292 with a gap of 1 mm, 211,768 with a gap of 2 mm, and 236,076 with a gap of 3 mm.

### 2.2. Material Properties

The base metal (BM) was pipeline steel, and the thermophysical parameters and mechanical properties of X80 are shown in Figure 3 [25]. The chemical composition of X80 is listed in Table 1. The microstructure consisted of a mixture of bainite ferrite and a small amount of M-A islands.

The filler material was ER70s-6, and the thermophysical parameters were set according to X80. The mechanical properties of ER70s-6 were tested by a tensile experiment based on ISO 6892-1:2019. The sample blank with a diameter of 10 mm was prepared using the surfacing method, and the tensile sample was taken from the surfacing sample, as shown in Figure 4. The tests were conducted at room temperature with a strain rate of 1 mm/min. The mechanical property parameters of ER70s-6 obtained through the tensile test are shown in Figure 5. The chemical composition of the filler material is shown in Table 2.

### 2.3. Heat Source Model

The Gaussian heat source model and double-ellipsoid heat source model are commonly used in welding numerical simulation. A Gauss heat source, belonging to the surface heat source model, is suitable for the simulation of thin-plate welding. The double-ellipsoid heat source model is often employed to simulate the deep penetration welding process of a medium and thick plate. Its shape parameters also need to be adjusted repeatedly to meet the actual welding pool shape and temperature field distribution. The characteristic of the homogeneous body heat source model is that the heat flux applied to the weld section is the same [26], which is ideal for multi-pass welding of a thick plate. Therefore, the homogeneous body heat source, i.e., where the heat is evenly distributed within a certain volume, was selected to conduct this investigation, which can be expressed as follows: (1)$$q=\frac{\eta UI}{V}$$
where *q* is the heat generation rate, *η* is the welding thermal efficiency, *U* is the arc voltage, *I* is the welding current, and *V* is the volume of the part where the heat source acts on.

The welding method applied in this study was multi-pass GMAW. In order to guarantee the deposition thickness of each pass, different assembly gaps usually need different welding heat inputs. So, the welding current was set to be 160 A (0 mm gap), 176 A (1 mm gap), 192 A (2 mm gap), and 208 A (3 mm gap). The arc voltage and welding speed were kept constant at 25 V and 50.8 cm/min, respectively. Based on the definition of heat input, the heat input of every pass of the welded joints was 376.5 J/mm (0 mm gap), 414.1 J/mm (1 mm gap), 451.8 J/mm (2 mm gap), and 489.4 J/mm (3 mm gap), respectively. 

### 2.4. Birth-Death Element and Boundary Conditions

#### 2.4.1. Temperature Field Model

For the temperature field model, the weld bead sequence was set by the birth-death element, and a complete multi-pass welding temperature field model was established. The control grid attribute is the heat transfer element. The thermal isolation boundary conditions were set on an axial symmetry plane, and the natural convection boundary conditions on both the inner surface and outer surface. Both the initial and inter-pass temperatures were set to 150 °C.

#### 2.4.2. Stress Field Model

The birth-death element setting was consistent with the temperature field model for the stress field model. Since the 1/8 arc model was adopted, it was necessary to better constrain the degree of freedom of the pipeline through cylindrical coordinates. In cylindrical coordinates, R represents radial degrees of freedom, T circumferential, and Z axial. The circumferential and axial degrees of freedom are constrained to T = 0 and Z = 0. The boundary conditions are shown in Figure 6.

The calculation of the welding residual stress field was mainly completed by introducing the temperature field. The heat transfer element was changed to a three-dimensional stress element. The grid of the temperature field and stress field were the same so as to avoid node interpolation calculation caused by grid inconsistency.

## 3. Results and Discussion

### 3.1. Temperature Field 

Figure 7a–g show the temperature field at t = 10 s of each pass in the welded joint without an assembly gap. The melting temperature of X80 steel was set to be 1500 °C. With the increase in the number of welding passes, the volume of the molten pool also expanded, and the heat influence range gradually expanded, although the heat input of each pass was uniform. This is the result of the heat superposition of each weld pass.

Figure 8, Figure 9 and Figure 10 illustrate the temperature field also at t = 10 s of each pass in the welded joint with 1 mm, 2 mm, and 3 mm assembly gap, respectively. The temperature distribution was broadly analogous to that when the butt gap was 0 mm, whereas the molten pool shape made a difference. The molten pool width increased with the increase in the butt gap. In fact, more metal was needed to be filled in the groove as the butt gap increased, and the wire-feeding speed must be increased accordingly, which meant that the welding heat input increased as for the GMAW process.

In order to verify the accuracy of the simulation results, the welded joint without a gap was selected as an example, and the thermal cycle curve was actually measured by a thermocouple. Figure 11a shows the thermal cycle comparison of the measured and simulated results at 5 mm away from the center of the cover weld. It can be seen that the measured and simulated curves were in good agreement. The measured t8/5 was 3.4 s, close to the simulated result (3.2 s). Figure 11b depicts the comparison between the simulated joint morphologies and the experimental ones. It is demonstrated that the simulation results are relatively accurate.

### 3.2. Welding Stress Distribution

#### 3.2.1. Residual Stress Field

Figure 12 shows the circumferential residual stress distribution with various butt gaps. It was noted that the maximum circumferential stress increased with the increase in the butt gap. The maximum circumferential stress (tensile stress) was 472.5 MPa when the assembly gap was up to 3 mm, 38.2% higher than the minimum (341.8 MPa) with no gap. The circumferential compressive stress (−270.5 MPa) with a 3 mm butt gap was also higher than that without gap (−76.3 MPa). This is related to the increase in heat input with the increase in the butt gap. Figure 13 displays the distribution curve of circumferential stress at different positions away from the weld center. The tensile stress was mainly distributed in the weld metal (WM) and heat-affected zone (HAZ), and the compressive stress was distributed in the area far from the weld center. The tensile–compressive transformation was completed around the HAZ. Von Mises stress is an equivalent stress considering the complex stress state in the object, which reflects the comprehensive stress state of the object. It was observed from Figure 14 that the equivalent stress in the WM was high, especially the root welding position, while the stress in the HAZ was relatively low. Additionally, the order of overall stress level under various butt gaps was: 3 mm > 2 mm > 1 mm > 0 mm. The highest equivalent stress was 467.3 MPa with a butt gap of 3 mm, which was 37.9% higher than that with no gap (338.9 MPa). The reason for the order of equivalent stress is that the increase in the butt gap leads to the increase in filler metal required for the weld, which accordingly brings about the increase in heat input and welding stress.

#### 3.2.2. Carrying Capacity

In the service process of transporting oil and gas, pipelines are required to bear certain internal pressure. The welded joint is needed to bear the combined action of internal pressure and residual stress. When the yield strength is reached, the pipe deforms, and when the tensile strength is reached, the pipe breaks. Therefore, it is generally believed that the internal pressure exerted when the material yields is the pressure limit that the pipeline can withstand. Through simulating the maximum equivalent stress (von Mises stress) of welded joints under different pipe internal pressures, Figure 15 was drawn, which describes the relationship between the maximum equivalent stress of a joint and internal pressure with various butt gaps. It was found that the maximum equivalent stress was positively correlated with internal pressure. With the increase in the butt gap, the internal pressure required to reach the yield strength (690 MPa) of the pipeline material, i.e., carrying capacity, decreased, as illustrated in Figure 16. The carrying capacity of the pipeline was only 13.1 MPa in regard to the welded joint with a butt gap of 3 mm. When no gap was reserved during welding, the pipeline carrying capacity achieved up to 17.8 MPa, 35.9 % higher than that with a 3 mm gap. 

The design internal pressure for X80 pipeline is 12.0 MPa. The pipeline carrying capacity is 15.2 MPa when the butt gap is 2 mm, so the safety margin is 3.2 MPa, which is generally deemed that the security is high. The smaller the butt gap, the higher the safety. Accordingly, the recommended butt gap should not exceed 2 mm during pipeline welding.

Figure 17 shows the equivalent stress distribution when the maximum reaches the material yield strength. For the joint without a butt gap, the maximum stress was concentrated in the WM. The distribution range of equivalent stress extended to the HAZ when the butt gap was reserved during welding. It can be deduced that plastic deformation or even fracture of the joint will occur in the corresponding area.

### 3.3. Mathematical Model of Butt Gap and Joint Carrying Capacity

According to the analysis of the joint carrying capacity *P* under different gap sizes *C* (Figure 18a), the mathematical model of two parameters was established, and the carrying capacity curve was fitted, as shown in Figure 18b.

The equation obtained by fitting with a univariate quadratic polynomial is as follows:(2)$$P=-0.125C^{2}-1.135C+17.715$$

Through this equation, the carrying capacity of the pipeline with other butt gaps can be predicted.

## 4. Conclusions

(1)Through the homogeneous body heat source model, the simulation results show a consistency with the thermocouple measurement outcomes. Temperature simulation results suggested that the width of weld pool and heat-affected zone (HAZ) increased along with the rise of the assembly gap due to the addition of heat input induced by increased filler metal.(2)The maximum circumferential stress of the joint raised with the increasing butt gap. The tensile stress, harmful to the joint performance, was mainly distributed in the weld metal and expanded to HAZ. The sequence of equivalent stress level under various butt gaps was: 3 mm > 2 mm > 1 mm > 0 mm, reaching the maximum of 467.3 MPa with a 3 mm butt gap, which was associated with the improvement in welding heat input as the gap increased.(3)The carrying capacity of the pipeline was positively correlated with the maximum equivalent stress of the joint, while there was a negative correlation between the carrying capacity and butt gap. The pipeline carrying capacity reached 17.8 MPa in regard to the joint without a butt gap, and dropped to 13.1 MPa for the joint with a 3 mm gap. The relationship between the carrying capacity (*P*) and butt gap (*C*) was expressed by *P* = −0.125*C*^2^ − 1.135*C* + 17.715, via which the pipeline carrying capacity with other butt gaps can be predicted.

## Figures and Tables

**Figure 1 materials-15-08299-f001:**
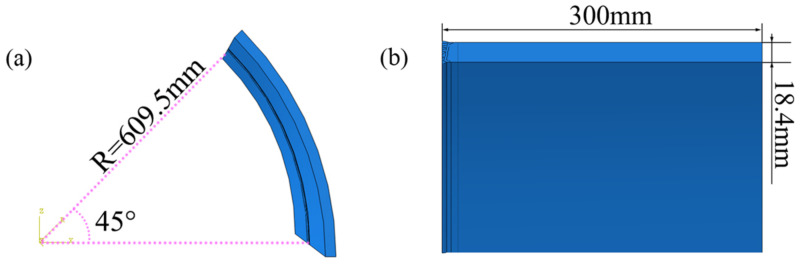
Specification of 1/8 geometric model of pipeline: (**a**) angle and radius, and (**b**) wall thickness and length of geometric model.

**Figure 2 materials-15-08299-f002:**
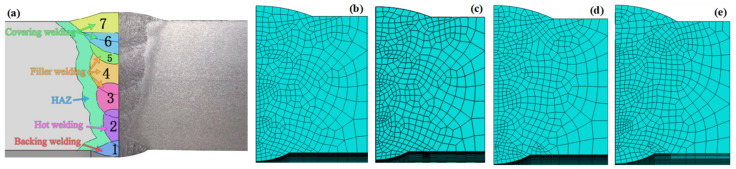
Geometric model of pipeline joint: (**a**) weld bead division, meshed model of (**b**) 0 mm, (**c**) 1 mm, (**d**) 2 mm, (**e**) 3 mm.

**Figure 3 materials-15-08299-f003:**
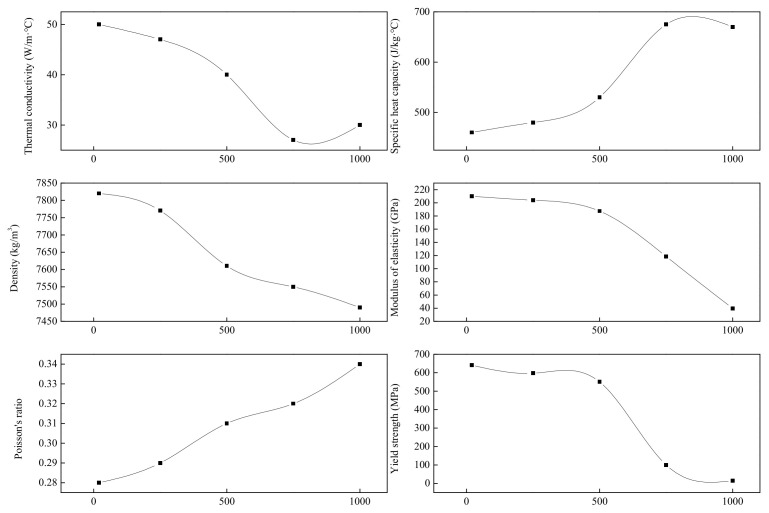
Thermophysical parameters and mechanical properties.

**Figure 4 materials-15-08299-f004:**
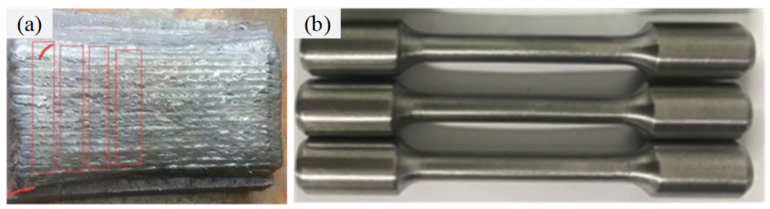
ER70s-6 properties test specimen: (**a**) surfacing sample, (**b**) tensile sample.

**Figure 5 materials-15-08299-f005:**
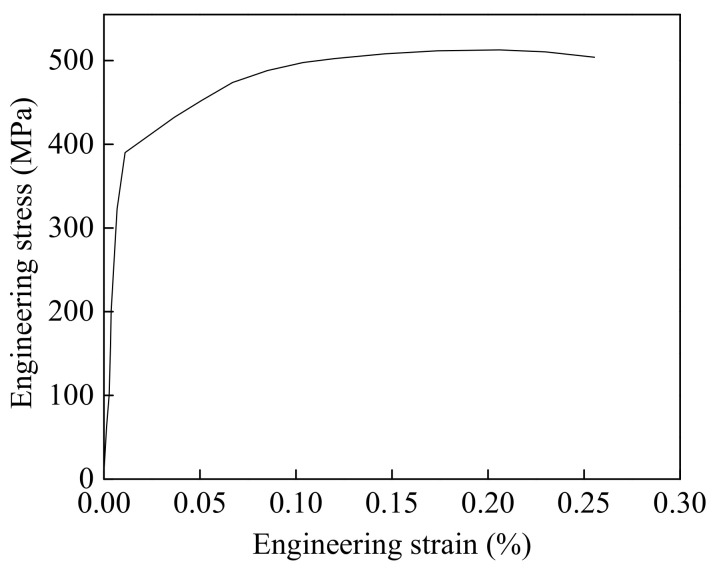
Stress–strain curve of ER70s-6.

**Figure 6 materials-15-08299-f006:**
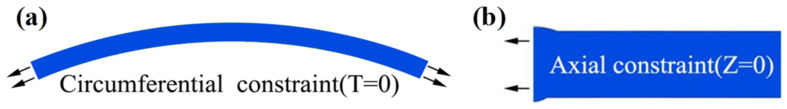
Stress field boundary conditions: (**a**) circumferential constraint, (**b**) axial constraint.

**Figure 7 materials-15-08299-f007:**
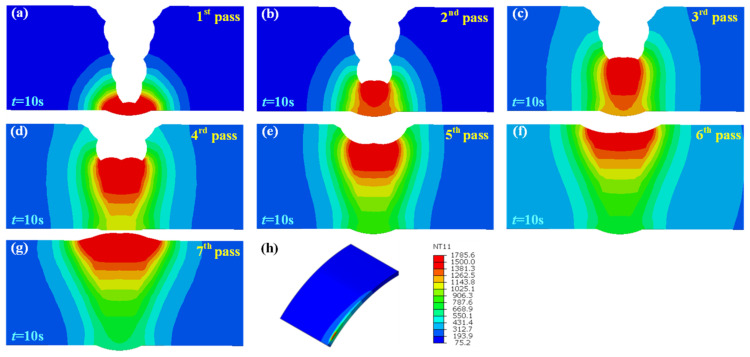
Joint temperature field with 0 mm butt gap: (**a**–**g**) temperature distribution of each pass, (**h**) isometric view of temperature field.

**Figure 8 materials-15-08299-f008:**
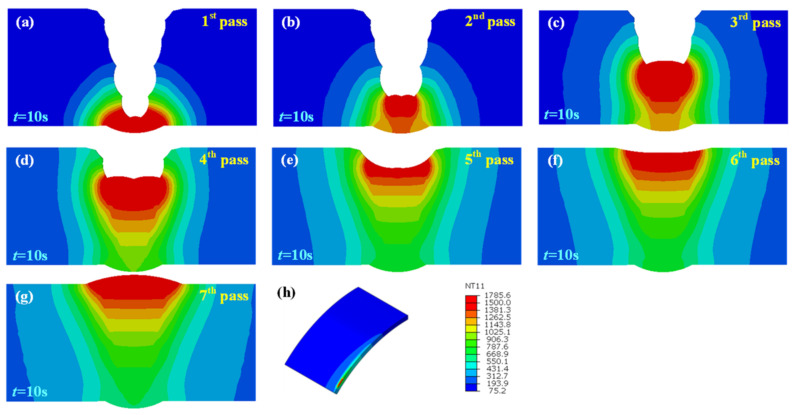
Joint temperature field with 1 mm butt gap: (**a**–**g**) temperature distribution of each pass, (**h**) isometric view of temperature field.

**Figure 9 materials-15-08299-f009:**
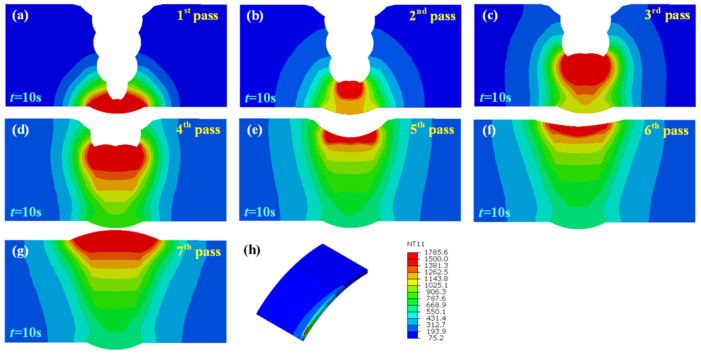
Joint temperature field with 2 mm butt gap: (**a**–**g**) temperature distribution of each pass, (**h**) isometric view of temperature field.

**Figure 10 materials-15-08299-f010:**
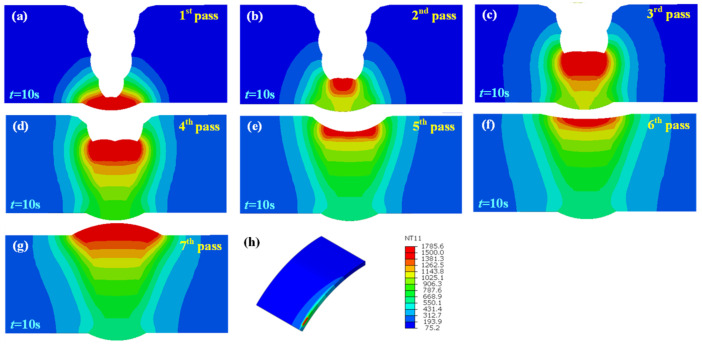
Joint temperature field with 3 mm butt gap: (**a**–**g**) temperature distribution of each pass, (**h**) isometric view of temperature field.

**Figure 11 materials-15-08299-f011:**
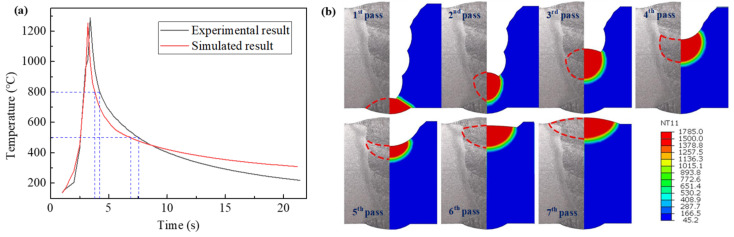
Comparison between simulation results and (**a**) thermal cycle measurement, (**b**) weld joint morphology.

**Figure 12 materials-15-08299-f012:**
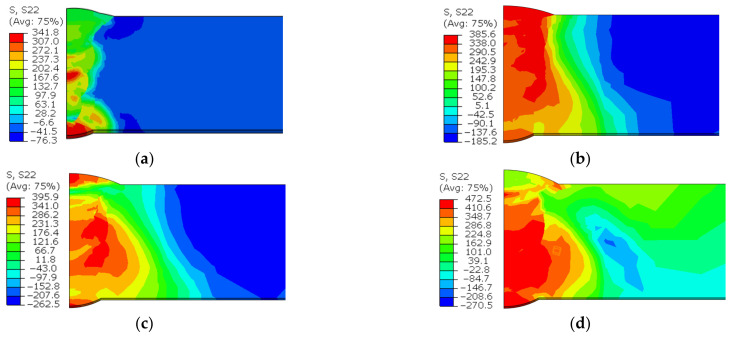
Circumferential stress nephogram with butt gaps of: (**a**) 0 mm, (**b**) 1 mm, (**c**) 2 mm, (**d**) 3 mm.

**Figure 13 materials-15-08299-f013:**
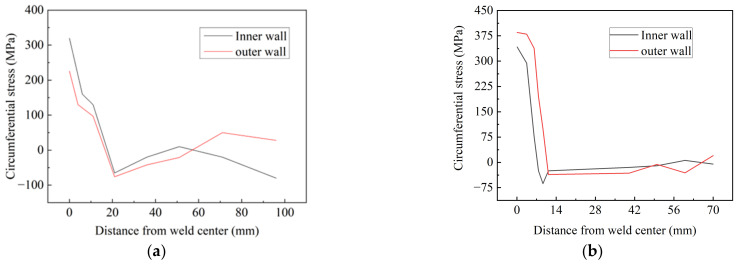
Circumferential stress distribution perpendicular to the weld direction with butt gaps of: (**a**) 0 mm, (**b**) 1 mm, (**c**) 2 mm, (**d**) 3 mm.

**Figure 14 materials-15-08299-f014:**
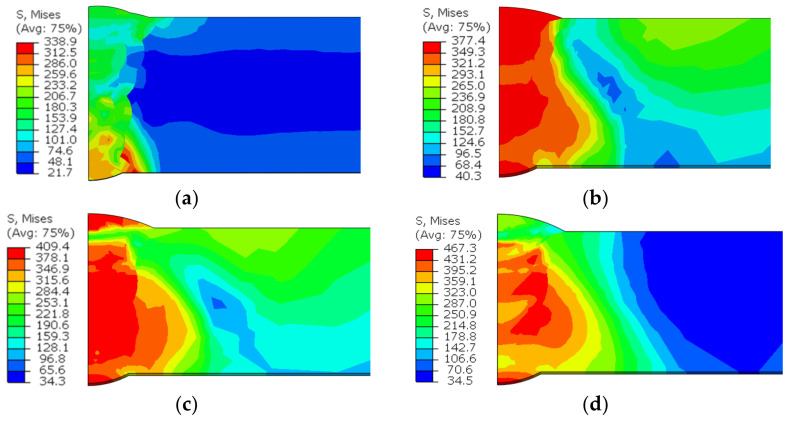
Equivalent stress field nephogram with butt gaps of: (**a**) 0 mm, (**b**) 1 mm, (**c**) 2 mm, (**d**) 3 mm.

**Figure 15 materials-15-08299-f015:**
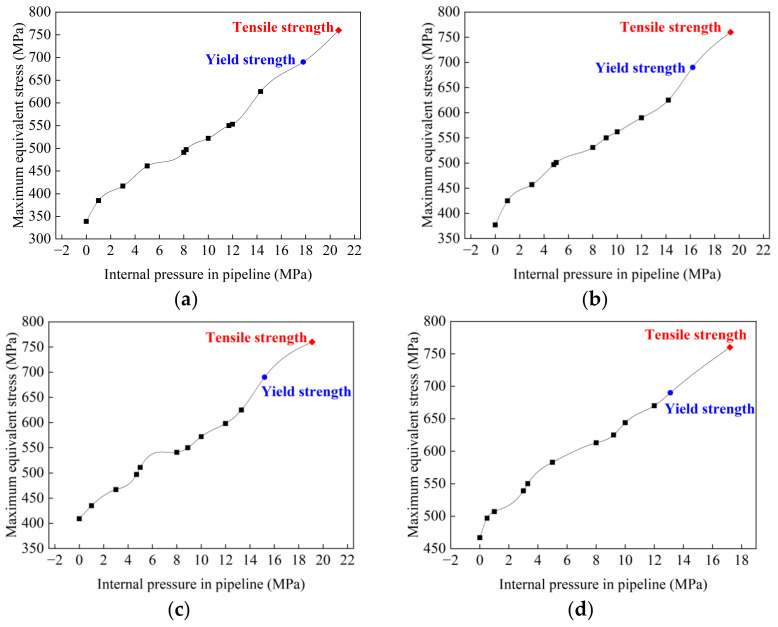
Relationship between joint equivalent stress and internal pressure with butt gap of: (**a**) 0 mm; (**b**) 1 mm; (**c**) 2 mm; (**d**) 3 mm.

**Figure 16 materials-15-08299-f016:**
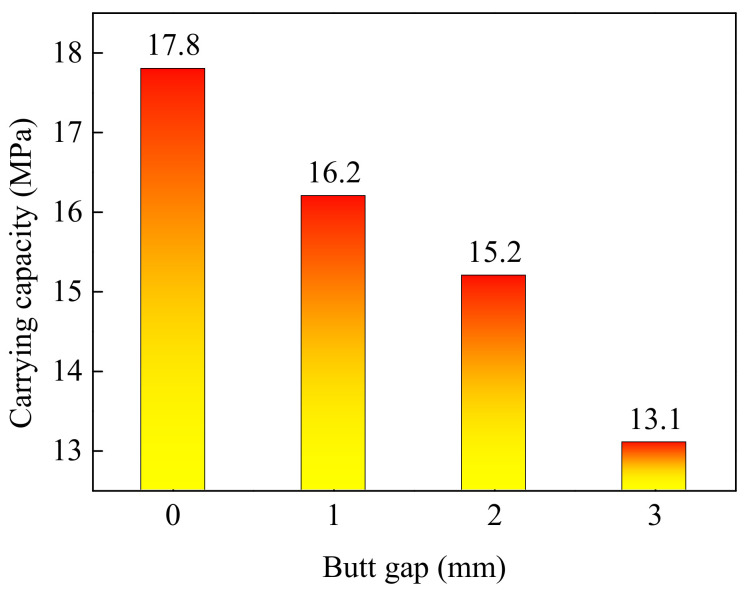
Carrying capability corresponding to different butt gaps.

**Figure 17 materials-15-08299-f017:**
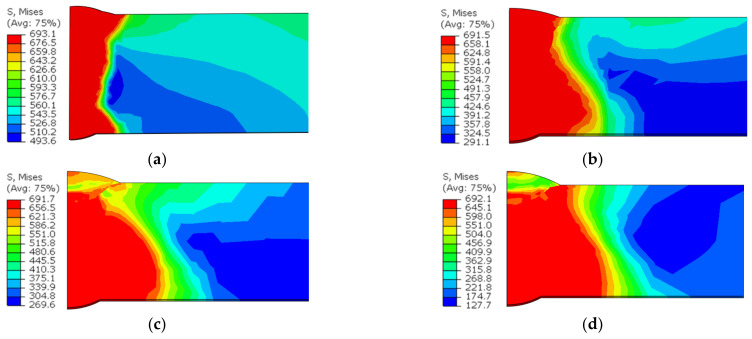
Equivalent stress distribution when the maximum reaches the material yield strength with butt gap of: (**a**) 0 mm; (**b**) 1 mm; (**c**) 2 mm; (**d**) 3 mm.

**Figure 18 materials-15-08299-f018:**
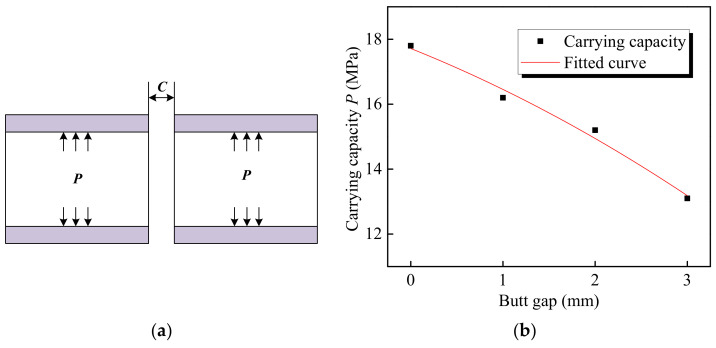
Relationship between carrying capacity and butt gap: (**a**) schematic diagram of pipeline bearing internal pressure; (**b**) curve fitting of carrying capacity.

**Table 1 materials-15-08299-t001:** Chemical composition of X80.

	C	Mn	Si	P	S	Cu	Ni	Cr	Nb	V	Al	Ti	Mo	Ca
X80	0.045	1.79	0.21	0.013	0.002	0.2	0.22	0.2	0.079	0.022	0.029	0.013	0.24	0.0025

**Table 2 materials-15-08299-t002:** Chemical composition of ER70s-6.

	C	Si	Mn	P≤	S≤	Ti	Ti + Zr	Al
ER70S-6	0.06	0.73	1.45	0.013	0.012	0.16	0.16	0.002

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
