# Peer review of "Effect of Butt Gap on Stress Distribution and Carrying Capacity of X80 Pipeline Girth Weld"

_materials, 2022, doi:10.3390/ma15238299_

Round 1
Reviewer 1 Report
This manuscript presented the “Effect of butt gap on stress distribution and carrying capacity of X80 pipeline girth weld”. The work considers the welding of X80 pipe with ER70s-6 with consideration of heat flow and residual stresses in different conditions. Also discussed and corelated with the mathematical model. This study is interesting to the engineering community since it can provide quantitative measure and data to support current and future applications of butt weld. However, there are plenty of information is missing and the manuscript needs revision and recheck before it can be accepted by the journal. The detailed comments and suggestions are listed as follows:
1.) Author has to give some key outcome in the abstract.
2.) Figure 1 shows two different views of pipe in one figure please shows separately by giving caption a and b.
3.) Figure 2 is not clear to understand please shows the details in figure such that weld pass, HAZ and bane material.
4.) Materials properties are calculated experimentally or considered from any reference. Please provide the details.
5.) Information is missing for the model such as numbers on nodes and elements, type of mash, type of element.
6.) What will be the maximum temperature in the model. Also add an isometric view of the model for thermal profile in the figure 7 to 10.
7.) From which method you get the thermal profile experimentally such as thermocouple measurement, infrared temperature measurement or anything else.
8.) Results of residual stress and carrying capacity are not discussed in text with consideration of values or percentage change in different condition.
9.) Re-write the conclusion part by including the main outcome.

Reviewer 2 Report
I believe that this draft successfully shows an application to determine the welding gap using computational welding analysis. Please, find the following comments.
1. Modeling procedure of finite element analysis.
I believe that authors have adopted the finite element analysis to obtain temperature and residual stress distribution.
Unfortunately, I'm afraid that this draft fails to describe the computational process and Finite element analysis in detail.
The analysis procedure including both FE model and Heat source model, should be explained in detail on the revised manuscript.
Please explain in detail the types of elements and boundary conditions, including the finite element analysis S/W adopted by authors. (The FE S/W has not been introduced in the manuscript.). The more descriptions of the FEA, the better contribution. Also, mesh shape & size used near the welding part should be introduced.
2. The heat source (mathematical model for heat flux or heat generation)
Section 2.3 and Equation (1) have described the welding heat source, but it is difficult to understand the heat source from them.
In welding analysis, the mathematical model of the heat source (i.e. Gaussian heat flux distribution, etc.) is the most important assumption and determine the result for analysis.
Also, I think that the process of evaluating whether the mathematical model of the heat source is appropriate (i.e. Comparing the HAZ areas measured by the experiment with those predicted by the analysis) must be addressed.
I hove the comments mentioned above will be reflected in the manuscript to be revised.
Reviewer 3 Report
The presented manuscript seems to be interesting for readers of the “Materials” journal, it is written in a good manner and suits the requirements of the journal. It can be accepted for publication after minor corrections listed below.
1- The dimensions and specifications of the model should be drawn on figure 1 and 2
2- The numbers on the vertical axis of Figure 3 in the Modulus of elasticity (GPa) versus time graph should be revised.
3- The chemical composition of X80 steel and filler material ER70s-6 should be presented in a table. Microscopic structure of X80 steel should be presented.
4- The dimensions of the samples and the tensile test standard and the specifications of the test (temperature and strain rate) should be provided
5- The clarity and quality of the images in Figure 6 should be improved.
6- The applied thermal cycle based on the distance from the welding center should be provided for several determined points. The temperature and time between passes should be provided.
7- Input heat for each butt gap should be provided. What number is considered for the melting temperature of steel? What criteria is the definition of the HAZ zone based on?
8- How to calculate and draw wind diagrams in figures 15, 16 and 19 are given in detail.
9- The results presented in the modeling section should be confirmed with practical evidence. The authors are requested to compare the practical results with the modeling results.
10- Based on the chemical composition and deposits in the base metal; The growth of austenite grain size during thermal cycling should be investigated. Please review and cite the following:
Materials Science and Technology, 30(4), 2014, 424-33.
11- Based on equivalent carbon, there is a possibility of martensite formation in the heat-affected zone of coarse grains. The growth of austenite grains and the increase of the probability of martensite formation affect the carrying capability and the strength of the connection, therefore, it is recommended that this issue be investigated by the authors. Doing this, review and citing the following ref could be helpful:
Neural Network World, 23, 2, 2013, 117.
12- Toughness and energy absorption to failure for different samples should be investigated and reported, and its relationship with structure and chemical composition should be investigated. For a more detailed review of the literature, I recommend referring to the following papers;
Neural Computing and Applications, 23, 2013, 2301-2308.
Journal of Mining and Metallurgy, Section B: Metallurgy, 51, 2015, 173-178.
13- Optimum sample should be specified. Is the cost of production checked for this selection?
Round 2
Reviewer 2 Report
I believe that the revised manuscript has well reflected the comments.
Reviewer 3 Report
As authors have performed an adequate revise, the manuscript might be accepted for publication in the journal of Materials.